# Learning Deep Graph Matching via Channel-Independent Embedding and Hungarian Attention

**Tianshu Yu[†], Runzhong Wang[‡], Junchi Yan[‡], Baoxin Li[†]**
[†]Arizona State University
[‡]Shanghai Jiao Tong University
`{tianshuy,baoxin.li}@asu.edu`
`{runzhong.wang,yanjunchi}@sjtu.edu.cn`

## Abstract

Graph matching aims to establishing node-wise correspondence between two graphs, which is a classic combinatorial problem and in general NP-complete. Until very recently, deep graph matching methods start to resort to deep networks to achieve unprecedented matching accuracy. Along this direction, this paper makes two complementary contributions which can also be reused as plugin in existing works: i) a novel node and edge embedding strategy which stimulates the multi-head strategy in attention models and allows the information in each channel to be merged independently. In contrast, only node embedding is accounted in previous works; ii) a general masking mechanism over the loss function is devised to improve the smoothness of objective learning for graph matching. Using Hungarian algorithm, it dynamically constructs a structured and sparsely connected layer, taking into account the most contributing matching pairs as hard attention. Our approach performs competitively, and can also improve state-of-the-art methods as plugin, regarding with matching accuracy on three public benchmarks.

## 1 Introduction

Without loss of generality, we consider the bijection problem for graph matching: given graph $\mathcal{G}_1$ and $\mathcal{G}_2$ of equal size $n$, graph matching seeks to find the one-vs-one node correspondence[1]:

$$\max_{\mathbf{x}} \mathbf{x}^\top \mathbf{K} \mathbf{x} \qquad \text{s.t.} \quad \mathbf{P} \mathbf{x} = \mathbf{1} \tag{1}$$

where $\mathbf{x} = \text{vec}(\mathbf{X}) \in \{0,1\}^{n^2}$ which is the column-wise vectorized form of the permutation matrix $\mathbf{X}$ that encodes the node-to-node correspondence between two graphs, and $\mathbf{K} \in \mathcal{R}_+^{n^2 \times n^2}$ is the so-called affinity matrix[2], respectively. Note $\mathbf{P}$ is a selection matrix encoding the one-to-one correspondence constraint. This problem is called Lawler's QAP (Lawler, 1963) and has attracted enormous attention for its generally NP-complete (Hartmanis, 1982) challenge, as well as a wide spectrum of applications in computer vision, graphics, machine learning and operational research etc. In particular, Koopmans-Beckmann's QAP (Loiola et al., 2007) with objective $\text{tr}(\mathbf{X}^\top \mathbf{F}_1 \mathbf{X} \mathbf{F}_2)$ is a special case of Eq. (1), which can be converted to Lawler's QAP by $\mathbf{K} = \mathbf{F}_2 \otimes \mathbf{F}_1$ and $\mathbf{F}_i$ refers to the weighted adjacency matrix. A series of solvers haven been developed to solve graph matching problem (Leordeanu & Hebert, 2005; Cho et al., 2010; Bernard et al., 2018; Yan et al., 2015; Yu et al., 2018). All these methods are based on deterministic optimization, which are conditioned with pre-defined affinity matrix and no learning paradigm is involved. This fact greatly limits the performance and broad application w.r.t. different problem settings considering its NP-hard nature.

Recently, the seminal work namely deep graph matching (DGM) (Zanfir & Sminchisescu, 2018) is proposed to exploit the high capacity of deep networks for graph matching, which achieves state-of-the-art performance. This is in contrast to some early works which incorporate learning strategy

---

[1]We assume graphs are of equal size for narrative simplicity. One can easily handle unbalanced graph size by adding dummy nodes as a common protocol in graph matching literature (Cho et al., 2010).

[2]$\mathbf{A}_{ia:jb}$ typically encodes the affinity between pair $(i,j)$ and $(a,b)$ where node $i,j \in \mathcal{G}_1$ and $a,b \in \mathcal{G}_2$.

separately in local stages (Caetano et al., 2009; Cho et al., 2013). On the other hand, Graph Convolutional Networks (GCN) (Kipf & Welling, 2017) brings about new capability on tasks over graph-like data, as it naturally integrates the intrinsic graph structure in a general updating rule:

$$\mathbf{H}^{(l+1)} = \sigma\left(\hat{\mathbf{A}}\mathbf{H}^{(l)}\mathbf{W}^{(l)}\right) \qquad (2)$$

where $\hat{\mathbf{A}}$ is the normalized connectivity matrix. $\mathbf{H}^{(l)}$ and $\mathbf{W}^{(l)}$ are the features and weights at layer $l$, respectively. Node embedding is updated by aggregation from 1-neighboring nodes, which is akin to the convolution operator in CNN. By taking advantages of both DGM and GCN, Wang et al. (2019) and Zhang & Lee (2019) incorporate permutation loss instead of displacement loss in (Zanfir & Sminchisescu, 2018), with notable improvement across both synthetic and real data.

Note that Eq. (1) involves both node and edge information, which exactly correspond to the diagonal and off-diagonal elements in $\mathbf{K}$, respectively. Edges can carry informative multi-dimensional attributes (namely weights) which are fundamental to *graph* matching. However existing embedding based graph matching methods (Wang et al., 2019; Xu et al., 2019) are focused on the explicit modeling of node level features, whereby the edges are only used as topological node connection for message passing in GCN. Besides, edge attributes are neither well modeled in the embedding-free model (Zanfir & Sminchisescu, 2018) since the edge information is derived from the concatenation of node features. To our best knowledge, there is no deep graph matching method explicitly incorporating edge attributes. In contrast, edge attributes e.g. length and orientation are widely used in traditional graph matching models (Cho et al., 2010; Yan et al., 2015; Yu et al., 2018) for constructing the affinity matrix $\mathbf{K}$. Such a gap shall be filled in the deep graph matching pipeline.

Another important consideration refers to the design of loss function. There are mainly two forms in existing deep graph matching works: i) displacement loss (Zanfir & Sminchisescu, 2018) similar to the use in optical flow estimation (Ren et al., 2017); ii) the so-called permutation loss (Wang et al., 2019) involving iterative Sinkhorn procedure followed by a cross-entropy loss. Results in (Wang et al., 2019) show the latter is an effective improvement against the former regression based loss. However, we argue that the continuous Sinkhorn procedure (in training stage) is yet an unnatural approximation to Hungarian sampling (in testing stage) for discretization. If the network is equipped with a continuous loss function (e.g. cross-entropy), we argue that the training process will make a great "meaningless effort" to enforce some network output digits of the final matching matrix into binary and neglect the resting digits which might have notable impact on accuracy.

This paper strikes an endeavor on the above two gaps and makes the following main contributions:

i) We propose a new approach for edge embedding via channel-wise operation, namely **channel-independent embedding** (**CIE**). The hope is to effectively explore the edge attribute and simulate the multi-head strategy in attention models (Veličković et al., 2018) by decoupling the calculations parallel and orthogonal to channel direction. In fact, edge attribute information has not been considered in existing embedding based graph matching methods (Wang et al., 2019; Xu et al., 2019).

ii) We devise a new mechanism to adjust the loss function based on the Hungarian method which is widely used for linear assignment problem, as termed by **Hungarian attention**. It resorts to dynamically generating sparse matching mask according to Hungarian sampling during training, rather than approximating Hungarian sampling with a differentiable function. As such, the Hungarian attention introduces higher smoothness against traditional loss functions to ease the training.

iii) The empirical results on three public benchmarks shows that the two proposed techniques are orthogonal and beneficial to existing techniques. Specifically, on the one hand, our CIE module can effectively boost the accuracy by exploring the edge attributes which otherwise are not considered in state-of-the-art deep graph matching methods; on the other hand, our Hungarian attention mechanism also shows generality and it is complementary to existing graph matching loss.

## 2   RELATED WORKS

**Graph embedding.** To handle graph-like data, early works adopt recursive neural networks (RNNs) treating input as directed acyclic graphs (Sperduti & Starita, 1997; Frasconi et al., 1998). Gori et al. (2005); Scarselli et al. (2008) generalized early models to graph neural networks (GNNs) so as to be directly applied on cyclic, directed or undirected graphs. Li et al. (2016) further improved this line

of model by replacing standard RNNs with gated recurrent units (GRUs) (Cho et al., 2013). Inspired by the great success of convolutional neural networks (CNNs) (Simonyan & Zisserman, 2014; He et al., 2016), researchers have made tremendous effort on applying convolution operator to graphs (Bruna et al., 2014; Kipf & Welling, 2017; Gong & Cheng, 2019). Bruna et al. (2014) defined a convolution operator in Fourier domain which is obtained by performing eigen-decomposition on graph Laplacian. However, such convolution will affect the whole spatial domain once taking inverse Fourier transformation. This method was improved by Chebyshev expansion to approximate filters (Defferrard et al., 2016). Kipf & Welling (2017) propose a graph convolutional operator over 1-neighbor nodes derived from graph spectral theory, which is invariant to node permutation and achieved significant performance on semi-supervised learning tasks. There are series of works following GCN, such as GraphSAGE (Hamilton et al., 2017), GAT (Veličković et al., 2018) and MPNN (Gilmer et al., 2017). Refer to (Cai et al., 2018) for a more comprehensive survey.

While the aforementioned models are focused on learning node state/embedding, a parallel line of work seek to learn edge embedding by taking into account the information carried on edges (Li et al., 2016; Gilmer et al., 2017; Gong & Cheng, 2019). Edges are intrinsic portion of graphs, and thus edge embedding can be essential to reveal the relation among nodes. Gilmer et al. (2017) introduce a general embedding network incorporating edge information and node-edge information merging, and a serious of works fall into this framework e.g. Gated GNN (Li et al., 2016), Tensor GNN (Schütt et al., 2017) and EGNN (Gong & Cheng, 2019). An improved version is devised in Chen et al. (2019) by interpreting this framework as maximizing mutual information across layers.

**Loss for combinatorial learning.** For the relatively easy linear assignment problem, it has been known that Sinkhorn algorithm (Sinkhorn, 1964) is the approximate and differentiable version of Hungarian algorithm (Mena et al., 2017). The Sinkhorn Network (Adams & Zemel, 2011) is developed given known assignment cost, whereby doubly-stochastic regulation is performed on input non-negative square matrix. Patrini et al. (2018) devise the Sinkhorn AutoEncoder to minimize Wasserstein distance, and Emami & Ranka (2018) propose to learning a linear assignment solver via reinforcement learning. For permutation prediction, DeepPermNet (Santa Cruz et al., 2018) adopts the Sinkhorn layer on top of a deep convolutional network. However this method cannot be directly applied for graph matching as it is not invariant to input permutations which is conditioned on a predefined node permutation as reference. In particular, existing supervised methods on combinatorial learning are generally cross-entropy-based. Pointer Net (Vinyals et al., 2015) incorporates cross-entropy loss on learning heuristics for combinatorial problems. Milan et al. (2017) propose an objective-based loss, where the gradients are only updated if the objective improves after update.

**Learning for graph matching.** The early effort (Caetano et al., 2009) aims to incorporate learning to graph matching. The key is to learn a more effective affinity function with given correspondence as supervision. While the ability by only learning affinity is limited, Cho et al. (2013) propose a matching function learning paradigm using histogram-based attributes with Structured-SVM (Tsochantaridis et al., 2005). A recent work (Zanfir & Sminchisescu, 2018) is a breakthrough to introduce deep learning paradigm into graph matching task, which utilizes a neural network to learn the affinity function. The learning procedure is explicitly derived from the factorization of affinity matrix (Zhou & De la Torre, 2012), which makes the interpretation of the network behavior possible. However, the displacement loss in (Zanfir & Sminchisescu, 2018) measures the pixel-wise translation which is similar to optical-flow (Dosovitskiy et al., 2015), being essentially a regression task instead of combinaotiral optimization. Seeing this limitation, Wang et al. (2019) employ element-wise binary cross-entropy, termed as permutation loss. This loss has proved capable of capturing the combinatorial nature rather than pixel offset, and achieves improvement over displacement loss. Node embedding is also used in (Wang et al., 2019) to explore the structure information.

## 3 THE PROPOSED LEARNING APPROACH FOR GRAPH MATCHING

### 3.1 APPROACH OVERVIEW

An overall structure of our approach is illustrated in Fig. 1. In line with (Wang et al., 2019), we employ VGG16 (Simonyan & Zisserman, 2014) to extract features from input images and bi-linearly interpolate the features at key points (provided by datasets). We concatenate lower-level (Relu4_2) and higher-level (Relu5_1) features to incorporate local and contextual information. For an image with $k$ key points, the feature is denoted as $\mathbf{H} \in \mathcal{R}^{k \times d}$, where $d$ is the feature dimension. Unless

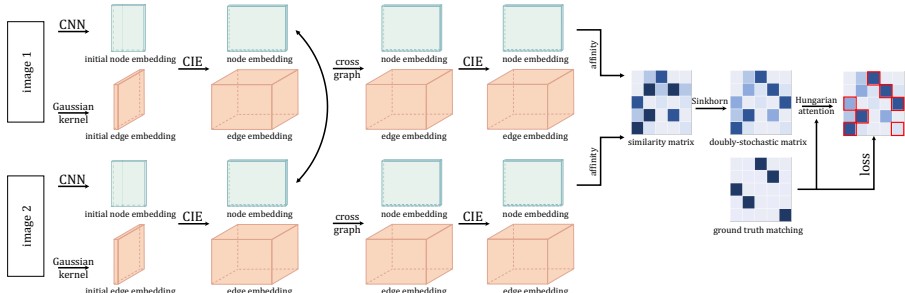

Figure 1: Architecture overview of the proposed deep graph matching networks that consist of the proposed channel-independent embedding and Hungarian attention layer over the loss function.

otherwise specified, the adjacency matrix $\mathbf{A} \in \mathcal{R}^{k \times k}$ is consequentially constructed via Delaunay triangulation (Delaunay et al., 1934), which is a widely adopted strategy to produce sparsely connected graph. To introduce more rich edge information, we also generate $k \times k$ $m$-dimensional edge features $\mathbf{E} \in \mathcal{R}^{m \times k \times k}$. $E$ can be initialized with some basic edge information (e.g. length and angle and other attributes) or a commutative function $\mathbf{E}_{ij} = p(\mathbf{H}_i, \mathbf{H}_j) = p(\mathbf{H}_j, \mathbf{H}_i) \in \mathcal{R}^m$, where $\mathbf{H}_i$ refers to the feature of node $i$. Note for directed graph, the commutative property is not required.

The features $\mathbf{H}$ and $\mathbf{E}$, together with the adjacency $\mathbf{A}$, are then fed into GNN module. Pairs of features are processed in a Siamese fashion (Bromley et al., 1994). Standard GCN's message passing rule simply updates node embedding as shown in Eq. (2). In contrast, each GNN layer in our model computes a new pair of node and edge embeddings simultaneously:

$$\mathbf{H}^{(l+1)} = f_i(\mathbf{H}^{(l)}, \mathbf{E}^{(l)}, \mathbf{A}; W_0^l), \quad \mathbf{E}^{(l+1)} = g(\mathbf{H}^{(l)}, \mathbf{E}^{(l)}, \mathbf{A}; W_1^l) \tag{3}$$

where $W_0^l$ and $W_1^l$ are the learnable parameters at layer $l$. The edge information is essential to provide structural feature enhancing graph matching. We initialize $\mathbf{H}^{(0)} = \mathbf{H}$ and $\mathbf{E}^{(0)} = \mathbf{E}$ in our setting. We will discuss the details of functions $f$ and $g$ in Sec. 3.2. Following state-of-the-art work (Wang et al., 2019), we also compute the cross-graph affinity followed by a column/row-wise softmax activation and a Sinkhorn layer (Adams & Zemel, 2011):

$$\mathbf{M}_{ij} = \exp\left(\tau \mathbf{H}_{(1)i}^\top \mathbf{\Lambda} \mathbf{H}_{(2)j}\right), \quad \mathbf{S} = \text{Sinkhorn}(\mathbf{M}) \tag{4}$$

Note here $\mathbf{M} \in \mathbb{R}^{k \times k}$ is the node-level similarity matrix encoding similarity between two graphs, differing from the edege-level affinity matrix $\mathbf{K}$ in Eq. 1. $\tau$ is the weighting parameter of similarity, $\mathbf{\Lambda}$ contains learnable parameters and $\mathbf{H}_{(1)i}$ is the node $i$'s embedding from graph $\mathcal{G}_1$. The output $\mathbf{S} \in [0, 1]^{k \times k}, \mathbf{S1} = \mathbf{1}, \mathbf{S}^\top \mathbf{1} = \mathbf{1}$ is a so-called doubly-stochastic matrix. Here $\text{Sinkhorn}(\cdot)$ denotes the following update iteratively to project $\mathbf{M}$ into doubly stochastic polygon:

$$\mathbf{M}^{(t+1)} = \mathbf{M}^{(t)} - \frac{1}{n}\mathbf{M}^{(t)}\mathbf{1}\mathbf{1}^\top - \frac{1}{n}\mathbf{1}\mathbf{1}^\top\mathbf{M}^{(t)} + \frac{1}{n^2}\mathbf{1}\mathbf{1}^\top\mathbf{M}^{(t)}\mathbf{1}\mathbf{1}^\top - \frac{1}{n}\mathbf{1}\mathbf{1}^\top \tag{5}$$

The Sinkhorn layer is shown to be an approximation of Hungarian algorithm which produces discrete matching output (Kuhn, 1955). As there are only matrix multiplication and normalization operators involved in Sinkhorn layer, it is differentiable. In practice, Eq. (5) converges rapidly within 10 iterations for decades of nodes. Less iterations involved, more precise back-propagated gradients can be achieved. We employ a cross-graph node embedding strategy following (Wang et al., 2019):

$$\mathbf{H}_{(1)}^{(l)} = f_c\left(\text{cat}(\mathbf{H}_{(1)}^{(l)}, \mathbf{S}\mathbf{H}_{(2)}^{(l)})\right), \quad \mathbf{H}_{(2)}^{(l)} = f_c\left(\text{cat}(\mathbf{H}_{(2)}^{(l)}, \mathbf{S}^\top\mathbf{H}_{(2)}^{(l)})\right) \tag{6}$$

where $f_c$ is a network and $\text{cat}(\cdot, \cdot)$ is the concatenation operator. $\mathbf{H}_{(i)}$ is the node feature of graph $i$. This procedure seeks to merge similar features from another graph into the node feature in current graph. It is similar to the feature transfer strategy in (Aberman et al., 2018) for sparse correspondence, which employs a feature merging method analogous to style transfer (Li et al., 2017).

As Sinkhorn layer does not necessarily output binary digits, we employ Hungarian algorithm (Kuhn, 1955) to discretize matching output $\mathbf{S}$ in testing. The testing differs from the training due to the Hungarian discretization. We introduce a novel attention-like mechanism termed as **Hungarian attention**, along with existing loss functions (will be detailed in Sec. 3.3). The final training loss is as follows, where $\mathbf{S}^G$ and $\mathcal{H}$ correspond to binary true matching and Hungarian attention loss.

$$\min \mathcal{H}(\mathbf{S}, \mathbf{S}^G) \tag{7}$$

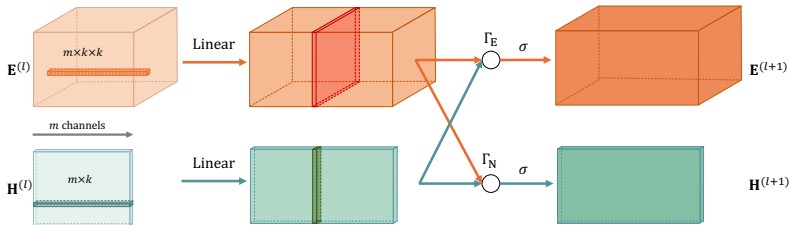

Figure 2: Illustration of the proposed CIE layer for embedding based deep graph matching. The operation "Linear" refers to the linear mapping, e.g. $\mathbf{H}_w^{(l)} \to \mathbf{W}_2^{(l)}\mathbf{H}_w^{(l)}$ in Eq (9).

## 3.2 CHANNEL-INDEPENDENT EMBEDDING

We detail the updating rule in Eq. (3). We propose a method to merge edge features into node features and perform matching on nodes. Edge information acts an important role in modeling relational data, whereby such relation can be complex thus should be encoded with high-dimensional feature. To this end, Gilmer et al. (2017) introduce a general embedding layer, which takes node and edge features and outputs a message to node $v$, then fuses the message and the current embedding:

$$\mathbf{m}_v^{(l)} = \sigma\left(\sum_{w\in\mathcal{N}_v} f_t\left(\mathbf{E}_{vw}\right)\mathbf{H}_w^{(l)} + \mathbf{W}^{(l)}\mathbf{H}^{(l)}\right), \quad \mathbf{H}_v^{(t+1)} = u_t\left(\mathbf{H}_v^{(t)}, \mathbf{m}_v^{(l)}\right) \tag{8}$$

where $\mathbf{E}_{vw}$ is the feature corresponding to edge $(v,w)$. In the realization of Eq. (8) (Gilmer et al., 2017), $\mathbf{m}_v^{(l)}$ and $\mathbf{H}_v^{(l)}$ are fed to GRU (Cho et al., 2014) as a sequential input. There are several variants which take into account specific tasks (Li et al., 2016; Schütt et al., 2017; Chen et al., 2019). Among these, Li et al. (2016) generates a transformation matrix for each edge and Schütt et al. (2017) resorts to merge embedding via fully connected neural networks. While edge-wise merging is straightforward, the representation ability is also limited. On the other hand, fully connected merging strategy will result in high computational cost and instability for back-propagation. To address these issues, we propose to merge embedding in a channel-wise fashion, which is termed as Channel-Independent Embedding (**CIE**). Concretely, the updating rule is written as:

$$\mathbf{H}_v^{(l+1)} = \sigma\left(\sum_{w\in\mathcal{N}_v} \underbrace{\Gamma_N\left(\mathbf{W}_1^{(l)}\mathbf{E}_{vw}^{(l)} \circ \mathbf{W}_2^{(l)}\mathbf{H}_w^{(l)}\right)}_{\text{channel-wise operator/function}}\right) + \sigma\left(\mathbf{W}_0^{(l)}\mathbf{H}_v^{(l)}\right) \tag{9}$$

$$\mathbf{E}_{vw}^{(l+1)} = \sigma\left(\mathbf{W}_1^{(l)}\mathbf{E}_{vw}^{(l)}\right) \tag{10}$$

where $\Gamma_N(\cdot \circ \cdot)$ is a channel-wise operator/function (above the underbrace), and it performs calculation per-channel and the output channel dimension is the same as input. The second $\sigma(\cdot)$ term is the message a node passes to itself, which is necessary in keeping the node information contextually consistent through each CIE layer. In this fashion, CIE is thus a procedure to aggregate node and edge embedding in each channel independently, which requires the dimensions of node ($\mathbf{W}_2^{(l)}\mathbf{H}_w^{(l)}$) and edge ($\mathbf{W}_1^{(l)}\mathbf{E}_{vw}^{(l)}$) representations to be equal. Similarly, we also propose an corresponding updating rule of edge embedding by substituting Eq. (10):

$$\mathbf{E}_{vw}^{(l+1)} = \sigma\left(\Gamma_E\left(\mathbf{W}_1^{(l)}\mathbf{E}_{vw}^{(l)} \circ h\left(\mathbf{H}_v^{(l)}, \mathbf{H}_w^{(l)}\right)\right)\right) + \sigma\left(\mathbf{W}_1^{(l)}\mathbf{E}_{vw}^{(l)}\right) \tag{11}$$

where $h(\cdot, \cdot)$ is commutative $h(\mathbf{X}, \mathbf{Y}) = h(\mathbf{Y}, \mathbf{X})$. Eq. (11) is supplementary to Eq. (9).

Fig. 2 shows a schematic diagram of CIE layer, which is motivated from two perspectives. First, CIE is motivated by counterparts in CNN (Qiu et al., 2017; Tran et al., 2018) which decouple a 3D convolution into two 2D ones (e.g. a $3 \times 3 \times 3$ convolution can be decomposed to a $1 \times 3 \times 3$ and a $3 \times 1 \times 1$ convolutions). In this sense, the number of parameters can be significantly reduced. As shown in Fig. 2, node and edge embedding is first manipulated along the channel direction via a linear layer, then operated via $\Gamma_N$ and $\Gamma_E$ orthogonal to the channel direction. Instead of merging node and edge as a whole, CIE layer decouples it into two operations. Second, CIE is also motivated by the triumph of multi-head structure (e.g. graph attention (Veličković et al., 2018)), the key of

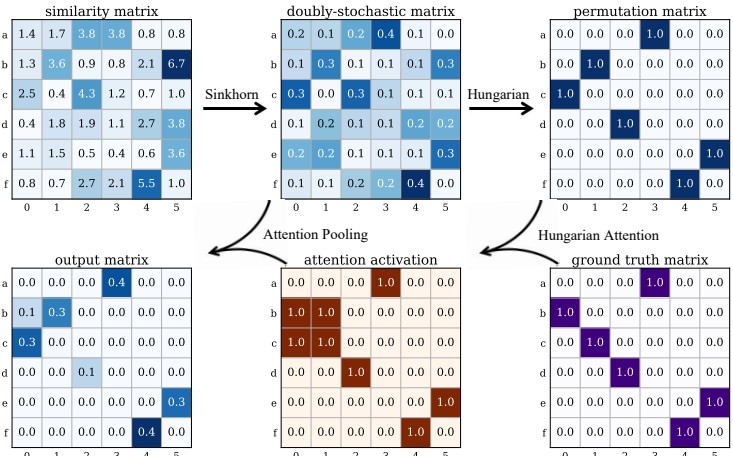

Figure 3: A working example illustrating of our proposed Hungarian attention pipeline starting from similarity matrix. Sinkhorn algorithm solves similarity matrix into a doubly-stochastic matrix in a differentiable way. A discrete permutation matrix is further obtained via Hungarian algorithm. Our proposed Hungarian attention, taking the ground truth matching matrix into account, focuses on the "important" digits either labeled true or being mis-classified. The output matrix is obtained by attention pooling from doubly-stochastic matrix, where we compute a loss on it.

which is to conduct unit calculation multiple times and concatenate the results. Multi-head proved effective to further improve the performance since it is capable of capturing information at different scales or aspects. Traditional neural node-edge message passing algorithms (Gilmer et al., 2017; Li et al., 2016; Schütt et al., 2017) typically produce a unified transformation matrix for all the channels. On the other hand, in Eq. (9) (10) and (11), one can consider that the basic operator in each channel is repeated $d$ times in a multi-head fashion. The cross-channel information exchange, as signified in Eq. (9) (10) and (11), only happens before the channel-wise operator (i.e. weights $\mathbf{W}_i^{(l)}$ as the cross-channel matrices). The main difference between CIE and traditional multi-head approaches e.g. (Veličković et al., 2018) is that CIE assumes the channel-independence of two embedded features (node and edge), while traditional ones only take one input under head-independence assumption.

### 3.3 HUNGARIAN ATTENTION MECHANISM

For most graph matching algorithms, the output is in a continuous domain. Though there are some alternatives that deliver discrete solutions by adding more constraints or introducing numerical continuation (Zhou & De la Torre, 2012; Yu et al., 2018), the main line of methods is to incorporate a sampling procedure (e.g. winner-take-all and Hungarian). Among them, the Hungarian algorithm (Kuhn, 1955) is a widely adopted, for its efficiency and theoretical optimality.

However, the Hungarian algorithm incurs a gap between training (loss function) and testing stages (Hungarian sampling). We compare the permutation loss (Wang et al., 2019) for concrete analysis:

$$\mathcal{L}_{\text{CE}} = - \sum_{i \in \mathcal{G}_1, j \in \mathcal{G}_2} \left( \mathbf{S}_{ij}^{\text{G}} \log \mathbf{S}_{ij} + \left(1 - \mathbf{S}_{ij}^{\text{G}}\right) \log\left(1 - \mathbf{S}_{ij}\right)\right) \tag{12}$$

Note Eq. (12) is an element-wise version of binary cross-entropy. During training, this loss tends to drag the digits in $\mathbf{S}$ into binary format and is likely trapped to local optima. This is because this loss will back-propagate the gradients of training samples that are easy to learn in the early training stage. In later iterations, this loss is then hard to give up the digits that have become binary. In fact, the similar phenomenon is also investigated in the focal loss (Lin et al., 2017) in comparison to the traditional cross-entropy loss. During the testing stage, however, the Hungarian algorithm has no preference on the case if digits in $\mathbf{S}$ are close to $0 - 1$ or not. It binarizes $\mathbf{S}$ anyway. Therefore, the effort of Eq. (12) to drag $\mathbf{S}$ into binary might be meaningless.

This issue is likely to be solved by integrating Hungarian algorithm during the training stage. Unfortunately, Hungarian algorithm is undifferentiable and its behavior is difficult to mimic with a differentiable counterpart. In this paper, instead of finding a continuous approximation of Hungarian algorithm, we treat it as a black box and dynamically generate network structure (sparse link)

Table 1: Accuracy on Pascal VOC (best in bold). White and gray background refer to results on **testing** and **training**, respectively. Compared methods include GMN (Zanfir & Sminchisescu, 2018), GAT (Veličković et al., 2018), EPN (Gong & Cheng, 2019), PCA/PIA (Wang et al., 2019).

| method | aero | bike | bird | boat | bottle | bus | car | cat | chair | cow | table | dog | horse | mbike | person | plant | sheep | sofa | train | tv | Ave |
|---|---|---|---|---|---|---|---|---|---|---|---|---|---|---|---|---|---|---|---|---|---|
| GMN-D | 31.9 | 47.2 | 51.9 | 40.8 | 68.7 | 72.2 | 53.6 | 52.8 | 34.6 | 48.6 | 72.3 | 47.7 | 54.8 | 51.0 | 38.6 | 75.1 | 49.5 | 45.0 | 83.0 | 86.3 | 55.3 |
| GMN-P | 31.1 | 46.2 | 58.2 | 45.9 | 70.6 | 76.4 | 61.2 | 61.7 | 35.5 | 53.7 | 58.9 | 57.5 | 56.9 | 49.3 | 34.1 | 77.5 | 57.1 | 53.6 | 83.2 | 88.6 | 57.9 |
| GAT-P | 46.4 | 60.5 | 60.9 | 51.8 | 79.0 | 70.9 | 62.7 | 70.1 | 39.7 | 63.9 | 66.2 | 63.8 | 65.8 | 62.8 | 39.5 | 82.0 | 66.9 | 50.1 | 78.5 | 90.3 | 63.6 |
| GAT-H | 47.2 | 61.6 | 63.2 | 53.3 | 79.7 | 70.1 | 65.3 | 70.5 | 38.4 | 64.7 | 62.9 | 65.1 | 66.2 | 62.5 | 41.1 | 78.8 | 67.1 | 61.6 | 81.4 | 91.0 | 64.6 |
| EPN-P | 47.6 | 65.2 | 62.2 | 52.7 | 77.8 | 69.5 | 63.4 | 69.6 | 37.8 | 62.8 | 63.6 | 63.9 | 64.6 | 61.9 | 39.9 | 80.5 | 66.7 | 45.5 | 77.6 | 90.6 | 63.2 |
| PIA-D | 39.7 | 57.7 | 58.6 | 47.2 | 74.0 | 74.5 | 62.1 | 66.6 | 33.6 | 61.7 | 65.4 | 58.0 | 67.1 | 58.9 | 41.9 | 77.7 | 64.7 | 50.5 | 81.8 | 89.9 | 61.6 |
| PIA-P | 41.5 | 55.8 | 60.9 | 51.9 | 75.0 | 75.8 | 59.6 | 65.2 | 33.3 | 65.9 | 62.8 | 62.7 | 67.7 | 62.1 | 42.9 | 80.2 | 64.3 | 59.5 | 82.7 | 90.1 | 63.0 |
| PCA-P | 40.9 | 55.0 | 65.8 | 47.9 | 76.9 | **77.9** | 63.5 | 67.4 | 33.7 | 65.5 | 63.6 | 61.3 | 68.9 | 62.8 | 44.9 | 77.5 | 67.4 | 57.5 | **86.7** | 90.9 | 63.8 |
| PCA-H | 49.8 | 60.7 | 63.9 | 52.6 | 79.8 | 72.5 | 63.8 | 71.2 | 38.4 | 62.5 | 71.7 | 65.4 | 66.6 | 62.5 | 40.5 | 84.7 | 66.1 | 47.9 | 80.5 | 91.1 | 64.6 |
| PCA+-P | 46.6 | 61.0 | 62.3 | 53.9 | 78.2 | 72.5 | 64.4 | 70.5 | 39.0 | 63.5 | **74.8** | 65.2 | 65.0 | 61.6 | 40.8 | 83.2 | 67.1 | 50.5 | 79.6 | 91.6 | 64.6 |
| $\text{CIE}_2$-P | 50.9 | 65.5 | 68.0 | 57.0 | 81.0 | 75.9 | 70.3 | 73.4 | **41.1** | 66.7 | 53.2 | 68.3 | 68.4 | 63.5 | 45.3 | 84.8 | 69.7 | 57.2 | 79.8 | 91.6 | 66.9 |
| $\text{CIE}_2$-H | 51.2 | 68.4 | 69.5 | 57.3 | 82.5 | 73.5 | 69.5 | 74.0 | 40.3 | 67.8 | 60.0 | 69.7 | 70.3 | 65.1 | 44.7 | 86.9 | 70.7 | 57.3 | 84.2 | 92.2 | 67.4 |
| $\text{CIE}_1$-P | **52.1** | **69.4** | 69.9 | **58.9** | 80.6 | 76.3 | **71.0** | **74.2** | **41.1** | 68.0 | 60.4 | 69.7 | 70.7 | 65.1 | **46.1** | 85.1 | **70.4** | 61.6 | 80.7 | 91.7 | 68.1 |
| $\text{CIE}_1$-H | 51.2 | 69.2 | **70.1** | 55.0 | **82.8** | 72.8 | 69.0 | **74.2** | 39.6 | **68.8** | 71.8 | **70.0** | **71.8** | **66.8** | 44.8 | **85.2** | 69.9 | **65.4** | 85.2 | **92.4** | **68.9** |
| PCA-P | 75.8 | 99.2 | 83.3 | 74.7 | 98.7 | 96.3 | 74.3 | 87.8 | 80.9 | 85.7 | 100.0 | 83.7 | 83.8 | 98.7 | 66.5 | 99.1 | 80.7 | 99.7 | 98.2 | 97.0 | 88.2 |
| $\text{CIE}_1$-P | 56.5 | 84.0 | 73.5 | 58.0 | 91.5 | 81.1 | 67.8 | 76.8 | 46.4 | 72.2 | 98.0 | 73.9 | 73.6 | 77.9 | 46.1 | 94.8 | 72.7 | 93.6 | 93.7 | 91.6 | 76.2 |
| $\text{CIE}_1$-H | 59.4 | 88.1 | 75.9 | 58.0 | 94.3 | 81.9 | 69.4 | 78.9 | 49.5 | 78.2 | 99.7 | 78.1 | 78.0 | 82.1 | 47.4 | 95.8 | 75.7 | 97.6 | 96.0 | 91.1 | 78.7 |

according to its output. Concretely, the sparse link is calculated as:

$$\mathbf{Z} = \text{Atten}\left(\text{Hungarian}(\mathbf{S}), \mathbf{S}^{\text{G}}\right) = \mathcal{P} \cup \mathcal{Q} \tag{13}$$

where the attention mechanism $\text{Atten}$ is fulfilled by an element-wise "logic OR" function. Fig. 3 shows an example of Hungarian attention procedure, and Eq. (13) highlights the most contributing digit locations: positive digits $\mathcal{P} = \mathbf{S}$ where Hungarian agrees with the ground-truth; negative digits $\mathcal{Q} = \text{Hungarian}(\mathbf{S}) \setminus \mathbf{S}^{\text{G}}$ where Hungarian differs from ground-truth. While GT (positive digits) naturally points out the digits that must be considered, negative ones indicate the digits that most hinder the matching (most impeding ones among all mis-matchings). Thus we need only minimize the loss at $\mathbf{Z}$, without considering the rest of digits. As we note that this mechanism only focuses on a small portion of the matching matrix which is analogous to producing hard attention, we term it **Hungarian attention**. Now that with the attention mask $\mathbf{Z}$, the Hungarian attention loss becomes:

$$\mathcal{H}_{\text{CE}} = - \sum_{i \in \mathcal{G}_1, j \in \mathcal{G}_2} \mathbf{Z}_{ij} \left(\mathbf{S}^{\text{G}}_{ij} \log \mathbf{S}_{ij} + \left(1 - \mathbf{S}^{\text{G}}_{ij}\right) \log\left(1 - \mathbf{S}_{ij}\right)\right) \tag{14}$$

Note that Hungarian attention mechanism can also be applied to other loss functions once the matching score is calculated in an element-wise fashion. Our experiment also studies Hungarian attention loss when casted on focal loss (Lin et al., 2017) and a specifically designed margin loss.

Finally we give a brief qualitative analysis on why Hungarian attention can improve matching loss. As discrete graph matching problem is actually built upon Delta function over permutation vertices (1 at ground-truth matching and 0 otherwise) (Yu et al., 2018), learning of graph matching with permutation loss is actually to approximate such functions with continuous counterparts. Unfortunately, more precise approximation to Delta function will result in higher non-smoothness, as discussed in Yu et al. (2018). For highly non-smooth objective, the network is more likely trapped at local optima. Hungarian attention, however, focuses on a small portion of the output locations, thus does not care about if most of the output digits are in $\{0, 1\}$. In this sense, Hungarian attention allows moderate smoothness of the objective, thus optimizer with momentum is likely to avoid local optima.

## 4 EXPERIMENTS

Experiments are conducted on three benchmarks widely used for learning-based graph matching: CUB2011 dataset (Welinder et al., 2010) following the protocol in (Choy et al., 2016), Pascal VOC keypoint matching (Everingham et al., 2010; Bourdev & Malik, 2009) which is challenging and Willow Object Class dataset (Cho et al., 2013). Mean matching accuracy is adopted for evaluation:

$$\text{Acc} = \frac{1}{k} \sum_{i \in \mathcal{G}_1, j \in \mathcal{G}_2} \text{AND}\left(\text{Hungarian}(\mathbf{S})_{ij}, \mathbf{S}^{\text{G}}_{ij}\right) \tag{15}$$

The algorithm abbreviation is in the form "X-Y", where "X" and "Y" refer to the network structure (e.g. CIE) and loss function (e.g. Hungarian attention loss), respectively. Specifically, **D**, **P** and **H**

correspond to displacement used in (Zanfir & Sminchisescu, 2018), permutation as adopted in (Wang et al., 2019) and Hungarian attention over permutation loss devised by this paper, respectively.

**Peer methods.** We compare our method with the following selected counterparts: 1) **HARG** (Cho et al., 2013). This shallow learning method is based on hand-crafted feature and Structured SVM; 2) **GMN** (Zanfir & Sminchisescu, 2018). This is a seminal work incorporating graph matching and deep learning, and the solver is upon spectral matching (Leordeanu & Hebert, 2005). While the loss of this method is displacement loss, we also report the results of GMN by replacing its loss with permutation loss (**GMN-P**); 3) **PIA/PCA** (Wang et al., 2019). PCA and PIA correspond to the algorithms with and without cross-graph node embedding, respectively. Readers are referred to Wang et al. (2019) for more details; We further replace the GNN layer in our framework with: 4) **GAT** (Veličković et al., 2018). Graph attention network is an attention mechanism on graphs, which reweights the embedding according to attention score; 5) **EPN** (Gong & Cheng, 2019). This method exploits multi-dimensional edge embedding and can further be applied on directed graphs. The edge dimension is set to 32 in our experiments. Finally, we term our network structure **CIE** for short. To investigate the capacity of edge embedding update, we also devise a version without edge embedding, in which connectivity is initialized as reciprocal of the edge length then normalized, rather than $\mathbf{A}$. This model is called **PCA+** since the node embedding strategy follows PCA.

**Implementation details.** As the node number of each graph might vary, we add dummy nodes for each graph pair such that the node number reaches the maximal graph size in a mini-batch in line with the protocol in (Wang et al., 2019). In either training or testing stages, these dummy nodes will not be updated or counted. The activation function in Eq. (9) (10) and (11) is set as Relu (Nair & Hinton, 2010) in all experiments. Specifically, the node and edge embedding is implemented by:

$$\mathbf{H}^{(l+1)}_{\cdot q} = \sigma\left(\left(\mathbf{A} \odot \left(\mathbf{W}^{(l)}_1 \mathbf{E}^{(l)}\right)_{\cdot q}\right)\left(\mathbf{W}^{(l)}_2 \mathbf{H}^{(l)}\right)_{\cdot q}\right) + \sigma\left(\left(\mathbf{W}^{(l)}_0 \mathbf{H}^{(l)}\right)_{\cdot q}\right) \tag{16a}$$

$$\mathbf{E}^{(l+1)}_{\cdot q} = \sigma\left(\left|\left(\mathbf{W}^{(l)}_0 \mathbf{H}^{(l)}\right)_{\cdot q} \ominus \left(\mathbf{W}^{(l)}_0 \mathbf{H}^{(l)}\right)^{\top}_{\cdot q}\right| \odot \mathbf{E}^{(l)}_{\cdot q}\right) + \sigma\left(\left(\mathbf{W}^{(l)}_1 \mathbf{E}^{(l)}\right)_{\cdot q}\right) \tag{16b}$$

where $\odot$ and $\ominus$ refer to element-wise product and pairwise difference, respectively. $\mathbf{H}_{\cdot q}$ is the $q$th channel of $\mathbf{H}$. In $\mathbf{CIE}_1$ setting, only node-level merging Eq. (16a) is considered and the edge feature is updated as Eq. (10). In $\mathbf{CIE}_2$ setting, we also replace the edge update Eq. (11) with Eq. (16b). Note edge embedding is used in both $\mathbf{CIE}_1$ and $\mathbf{CIE}_2$ and note **PCA-H** can be regarded as the pure node embedding version of our approach. The edge feature is initiated as reciprocal of the edge length. For training, batch size is set to $8$. We employ SGD optimizer (Bottou, 2010) with momentum 0.9. Two CIE layers are stacked after VGG16.

**CUB2011 test** CUB2011 consists of 11,788 images from 200 kinds of birds with 15 annotated parts. We randomly sample image pairs from the dataset following the implementation released by Choy et al. (2016). We do not use the pre-alignment of poses during testing, because their alignment result is not publicly available. Therefore, there exists significant variation in pose, articulation and appearance across images, in both training and testing phase. Images are cropped around bounding box and resized to $256 \times 256$ before fed into the network. Instead of evaluating the performance in a retrieval fashion (Zanfir & Sminchisescu, 2018), we directly evaluate the matching accuracy since the semantic key-points are pre-given. We test two settings: 1) *intra-class*. During training, we randomly sample images, with each pair sampled from the same category (out of 200 bird categories). In testing, 2,000 image pairs (100 pairs for each category) are sampled; 2) *cross-class*. We analogously sample image pairs without considering the category information and 5,000 randomly sampled image pairs are employed for testing. While the first setting is for a class-aware situation, the second setting is considered for testing the class-agnostic case. Results are shown in Table 3.

We see our method surpasses all the competing methods in terms of matching accuracy. Besides, almost all the selected algorithms can reach over $90\%$ accuracy, indicating that this dataset contains mostly "easy" learning samples. In this case, the Hungarian attention can slightly improve the performance since easy gradients agree with descending trend of the loss on the whole dataset.

**Pascal VOC test** The Pascal VOC dataset with Key-point annotation (Bourdev & Malik, 2009) contains 7,020 training images and 1,682 testing images with 20 classes in total. To the best of our knowledge, this is the largest and most challenging dataset for graph matching in computer vision. Each image is cropped around its object bounding box and is resized to $256 \times 256$. The node

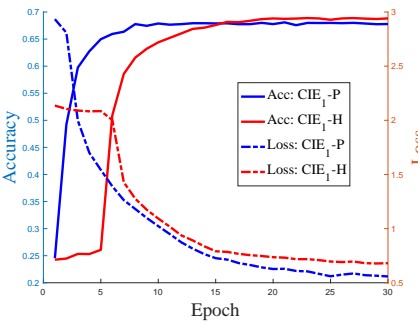
(a) Accuracy/loss vs. training epoch.

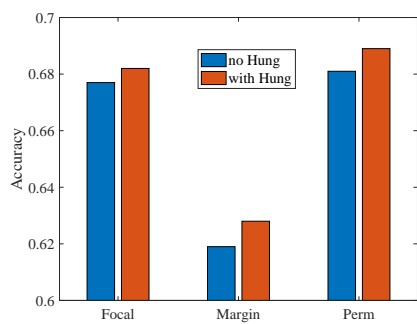
(b) Ablation study by Hungarian attention.

Figure 4: Performance study on Pascal VOC. Note in (a) the loss is calculated on all matching digits for both CIE$_1$-P and CIE$_1$-H. Note around 10th epoch, the accuracy of CIE$_1$-P almost reaches the highest, but the loss keeps descending until 30th epoch. This indicates that in most of the latter epochs, P-loss performs "meaningless" back-propagation to drag the output to binary. H-loss, by accommodating smoothness, can emphasize most contributing digits and achieves higher accuracy.

size of this dataset varies from 6 to 23 and there are various scale, pose and illumination perturbations. Experimental results are summarized in Table 1. We see in either setting, CIE significantly outperforms all peer algorithms. Specifically, CIE$_1$-H achieves the best performance and has $0.8\%$ improvement w.r.t. average accuracy over CIE$_1$-P. For each class, CIE$_1$-H and CIE$_1$-P carve up most of the top performance. We also note that CIE$_1$-H has a close performance on "table" compared with GMN-D. Since P-loss is naturally not as robust as D-loss on symmetric objects, P-loss showed great degradation over D-loss on "table" (as discussed in (Wang et al., 2019)). However, with the help of Hungarian link, H-loss can maintain relatively high accuracy despite natural flaw of P-loss. This observation indicates that H-loss can focus on "difficult" examples. We also note that CIE$_1$ produces better results against CIE$_2$, which implies that updating edge embedding is less effective compared to a singleton node updating strategy. We can also see from Table 1 that PCA-P has much higher performance on *training samples* than CIE$_1$-H, which is to the contrary of the result on testing samples. This might indicate that PCA-P overfits the training samples.

**Accuracy/loss vs. training epoch.** We further show the typical training behavior of P-loss and H-loss on Pascal VOC dataset in Fig. 4. 30 epochs are involved in a whole training process. Accuracy is evaluated on *testing samples* after each epoch while loss is the average loss value within each epoch. In the early training stage, the loss of CIE$_1$-P immediately drops. On the other hand, CIE$_1$-H hesitates for several epochs to find the most effective descending direction. On the late stage, we observe that even though P-loss (Eq. (12)) calculates much more digits than H-loss (Eq. (14)), the loss values are opposite. This counter-intuitive fact strongly indicates that P-loss makes meaningless effort, which is not helpful to improve the performance, at late stage. The proposed H-loss, on the other hand, is capable of avoiding easy but meaningless gradients.

**Effect of Hungarian attention mechanism.** We also conduct experiments to show the improvement of Hungarian attention over several loss functions (with and without Hungarian attention): Hungarian attention is applied on Focal loss (Focal) (Lin et al., 2017) as:

$$\mathcal{L}_{\text{focal}} = \begin{cases} -\alpha \mathbf{Z}_{ij}(1 - \mathbf{S}_{ij})^\gamma \log(\mathbf{S}_{ij}), & \mathbf{S}_{ij}^{\text{G}} = 1 \\ -(1 - \alpha)\mathbf{Z}_{ij}\mathbf{S}_{ij}^\gamma \log(1 - \mathbf{S}_{ij}), & \mathbf{S}_{ij}^{\text{G}} = 0 \end{cases} \tag{17}$$

where controlling parameters $\alpha = 0.75$ and $\gamma = 2$ in our setting. We also design a margin loss (Margin) with Hungarian attention under a max-margin rule. Note we insert the Hungarian attention mask $\mathbf{Z}_{ij}$ into Eq. (17) and Eq. (18) based on the vanilla forms.

$$\mathcal{L}_{\text{margin}} = \begin{cases} \mathbf{Z}_{ij} \times \max(1 - \mathbf{S}_{ij} - \beta, 0), & \mathbf{S}_{ij}^{\text{G}} = 1 \\ \mathbf{Z}_{ij} \times \max(\mathbf{S}_{ij} - \beta, 0), & \mathbf{S}_{ij}^{\text{G}} = 0 \end{cases} \tag{18}$$

where we set the margin value $\beta = 0.2$. Loss of Eq. (18) is valid because after Softmax and Sinkhorn operations, $\mathbf{S}_{ij} \in [0, 1]$. We also show permutation loss (Perm) (Wang et al., 2019). Result can be found in Fig. 4 (b) whereby the average accuracy on Pascal VOC is reported. All the settings are under CIE$_1$. For either loss, the proposed Hungarian attention can further enhance the accuracy, which is further visualized by a pair of matching results under P-loss and H-loss in Fig. 5.

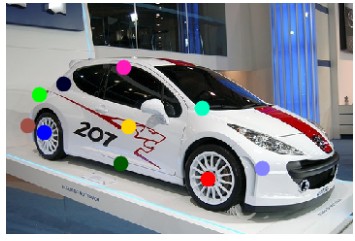 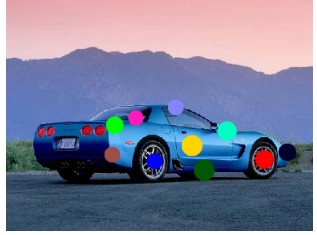 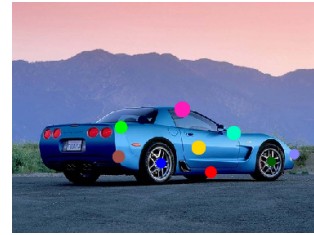

(a) Reference Image (b) P-loss: **7/10** (c) H-loss: **8/10**

Figure 5: Visualization of a matching result: 10 key points in each image with 7 and 8 correct matchings dispalyed, respectively. Different colors across images indicate node correspondence. The larger size of dot, the larger is the predicted value $\mathbf{S}_{ij}$. (a) The reference image. (b) Result on the target image from $CIE_1$-P. (c) Result on the target image from $CIE_1$-H. We see though H-loss i.e. Hungarian attention loss outputs smaller predicted values, it delivers a more accurate matching.

Table 2: Accuracy (%) on Willow Object.

| method | face | mbike | car | duck | wbottle |
|--------|------|-------|-----|------|---------|
| HARG | 91.2 | 44.4 | 58.4 | 55.2 | 66.6 |
| GMN-V | 98.1 | 65.0 | 72.9 | 74.3 | 70.5 |
| GMN-W | 99.3 | 71.4 | 74.3 | 82.8 | 76.7 |
| PCA-V | **100.0** | 69.8 | 78.6 | 82.4 | 95.1 |
| PCA-W | **100.0** | 76.7 | **84.0** | **93.5** | 96.9 |
| CIE-V | 99.9 | 71.5 | 75.4 | 73.2 | **97.6** |
| CIE-W | **100.0** | **90.0** | 82.2 | 81.2 | **97.6** |

Table 3: Accuracy (%) on CUB.

| method | intra-class | cross-class |
|--------|-------------|-------------|
| GMN-D | 89.6 | 89.9 |
| GMN-P | 90.4 | 90.8 |
| GAT-P | 93.2 | 93.4 |
| PCA-P | 92.9 | 93.5 |
| PCA-H | 93.7 | 93.5 |
| CIE-P | 94.1 | 93.8 |
| CIE-H | **94.4** | **94.2** |

**Willow Object Class test** We test the transfer ability on Willow Object Class (Cho et al., 2013). It contains 256 images[3] of 5 categories in total, with three categories (face, duck and winebottle) collected from Caltech-256 and resting two (car and motorbike) from Pascal VOC 2007. This dataset is considered to have bias compared with Pascal VOC since images in the same category are with relatively fixed pose and background is much cleaner. We crop the object inside its bounding box and resize it to $256 \times 256$ as CNN input. While HARG is trained from scratch following the protocol in (Cho et al., 2013), all the resting counterparts are either directly pre-trained from the previous section or fine-tuned upon the pre-trained models. We term the method "X-**V**" or "X-**W**" to indicate pre-trained model on Pascal **V**OC or fine-tuned on **W**illow, respectively. CIE refers to $CIE_1$-H for short. Results in Table 2 suggest that our method is competitive to state-of-the-art.

## 5 CONCLUSION

We have presented a novel and effective approach for learning based graph matching. On one hand, the novelty of our method partially lies in the development of the Hungarian attention, which intrinsically adapts the matching problem. It is further observed from the experiments that Hungarian attention can improve several matching-oriented loss functions, which might bring about potential for a series of combinatorial problems. On the other hand, we also devise the channel independent embedding (CIE) technique for deep graph matching, which decouples the basic merging operations and is shown robust in learning effective graph representation. Extensive experimental results on multiple matching benchmarks show the leading performance of our solver, and highlight the orthogonal contribution of the two proposed components on top of existing techniques.

ACKNOWLEDGMENTS

Tianshu Yu and Baoxin Li were supported in part by a grant from ONR. Any opinions expressed in this material are those of the authors and do not necessarily reflect the views of ONR. Runzhong Wang and Junchi Yan were supported in part by NSFC 61972250 and U19B2035.

---

[3]The data size is too small to train a deep model. Hence we only evaluate the transfer ability on this dataset.

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

## A APPENDIX

### A.1 SYNTHETIC TEST

Synthetic graphs are generated for training and testing following the protocol in (Cho et al., 2010). Specifically, $K_{pt}$ keypoints are generated for a pair of graphs with a 1024-dimensional random feature for each node, which is sampled from uniform distribution $\mathcal{U}(-1, 1)$. Disturbance is also applied to graph pairs including: Gaussian node feature noise from $\mathcal{N}(0, \sigma_{ft}^2)$; random affine transformation $\begin{bmatrix} s\cos\theta & -s\sin\theta & t_x \\ s\sin\theta & s\cos\theta & t_y \\ 0 & 0 & 1 \end{bmatrix}$ with $s \sim \mathcal{U}(0.8, 1.2), \theta \sim \mathcal{U}(-60, 60), t_x, t_y \sim \mathcal{U}(-10, 10)$ followed by Gaussian coordinate position noise $\mathcal{N}(0, \sigma_{co}^2)$. By default we assign $K_{pt} = 25, \sigma_{ft} = 1.5, \sigma_{co} = 5$. Two graphs share the same structure. We generate 10 random distributions for each test. Results are shown in Fig. 6. The performance of PCA and CIE is reported. We see our method significantly outperformed PCA. It can further be noticed that Hungarian attention can help to achieve an even higher accuracy. Readers are referred to Wang et al. (2019) for some other results on synthetic test.

However, we also notice that the way to generate synthetic graphs is much different from the distribution of real-world data. For real-world data, on one hand, there is strong correlation on the

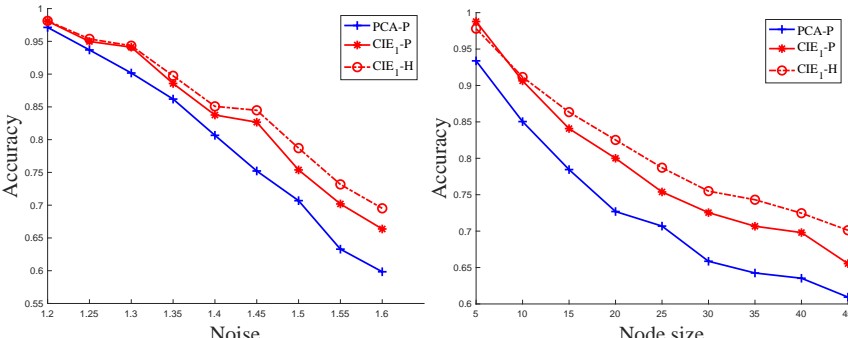

Figure 6: Results on synthetic test where two different loss functions are compared in ablative study.

neighboring node features. This is the reason why the message passing from nearby node features works. However, the features of synthetic data are randomly generated and there is no correlation between neighboring node features. Therefore, message passing mechanism is not very effective to reveal the relation or pattern among local nodes for synthetic data. On the other hand, features of real-world data typically lie on a manifold embedded in high dimensional space, hence is low dimensional. However, randomly generated features will span the whole space and show no patterns.

Taking into account the aforementioned factors, we believe there is a demand for a novel strategy to generate more reasonable synthetic data. This can be one of the future works.

## A.2 COMPARISON OF PASCAL VOC AND WILLOW

As we claim that Willow dataset is biased compared with Pascal VOC dataset, we qualitatively show some randomly selected examples in Fig. 7. We select several images with the same class "car" from both datasets. We also choose images with "bird" from Pascal VOC and "duck" from Willow since they somewhat share similar semantic information. We see in either case, Pascal VOC contains more variation and degradation compared with Willow in terms of pose, scale, appearance, etc. In general, Willow dataset is easier for algorithms to learn. While there is a significant performance gap of PCA over these two datasets, the performance of CIE on Willow without fine-tune (Table 2) is consistent to the performance on Pascal VOC (Table 1). As such, we infer the performance degradation of CIE on "duck" in Willow test (Table 2) is due to such bias. The pre-trained CIE on Pascal VOC tends to produce more stable and higher average accuracy on *all types of images*, rather than focusing on "easy-to-learn" samples by PCA. This is a different learning strategy from PCA.

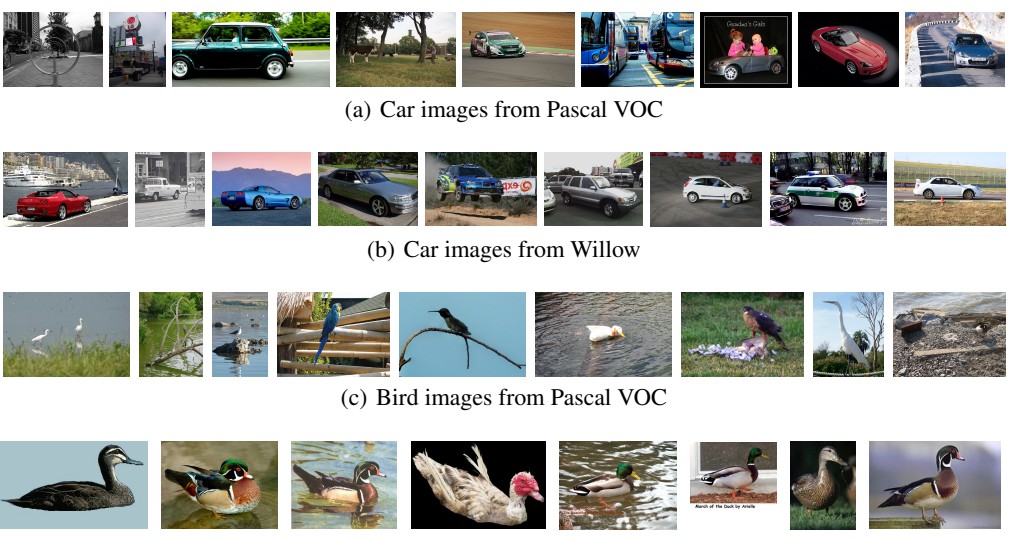

(a) Car images from Pascal VOC

(b) Car images from Willow

(c) Bird images from Pascal VOC

(d) Duck images from Willow

Figure 7: Image examples from Pascal VOC and Willow.

