# OpenReview forum: "Learning deep graph matching with channel-independent embedding and Hungarian attention"
_ICLR.cc/2020/Conference — Accept (Poster)_

### Official Review · AnonReviewer2 · 2019-10-23
**Official Blind Review #2**

**Rating:** 3

**Review:**

This paper studies the graph matching problem in the context of vision. Although I am familiar with the graph matching problem, I have much less experience regarding its application in vision. My understanding is that features are extracted from images and used to construct a graph. This graph is then passed through a GNN but there are steps which are unclear. In particular on page 5, 'm and H are fed to a GRU as a sequential input' but I do not see where this fit into the architecture. Similarly, I am a bit confused by equations (8) and (9) as there is a H^(t+1)_v in both equations. Does equation (9) make the function u_t explicit? Then what is \Gamma_N and similarly what is \Gamma_E in equation (11)?
As it is written, this paper seems more appropriate for a conference in vision.

**Experience Assessment:**

I do not know much about this area.

**Review Assessment: Checking Correctness Of Derivations And Theory:**

N/A

**Review Assessment: Checking Correctness Of Experiments:**

I did not assess the experiments.

**Review Assessment: Thoroughness In Paper Reading:**

I read the paper at least twice and used my best judgement in assessing the paper.

---

> ### Author Response · Authors · 2019-11-13
> **response to review#2**
>
> Thanks for your specific questions that will help improve our paper. To clarify, Eq (8) corresponds to the GNN update rule in [Gilmer et al. 2017] (in which the read-out function is a GRU) and Eq (9) is our update rule (CIE). So Eq (8) and (9) are for different methods. We do not employ any GRUs in our algorithm. Instead, we adopt a Graph Convolutional Layer with ReLU activation.
>
> Our main contribution is to propose a Hungarian attention mechanism, which dynamically generates links in computational graph and proved beneficial to graph matching. Though the experiments are confined in vision for the relatively standard experiment protocol in this area and rich public benchmark, while as discussed in our response to review#4, graph matching itself is a fundamental problem in computer science.
>
> And more importantly, the added new results in response to review#1 (imposing Hungarian attention to [1])  showed that our Hungarian attention is a general and effective means to improve graph matching (in [1] geometric edge features are utilized). In fact, the paper is written in an abstractive level in terms of the technical approach (the edge embedding and Hungarian attention are both general and have nothing to do with images), and the related work. Though the experiments are concerned with vision setting for the reasons we mention above (and also due to space limitation). We will add more broad discussion in the introduction and add the new results in the experiment part.
>
> [1] Zhen Zhang, and Wee Sun Lee. "Deep Graphical Feature Learning for the Feature Matching Problem.", ICCV 2019

---

### Official Review · AnonReviewer1 · 2019-10-31
**Official Blind Review #1**

**Rating:** 6

**Review:**

The authors proposed a new way to train graph siamese networks for the graph matching problem. The overall framework of this paper is somehow similar to [1] and [2], except for the final Hungarian attention module, which is the key contribution of this paper. In the current settings, the authors proved that using their Hungarian attention module, the performance can be improved. However, it would be better if the Hungarian attention can be applied to DGCNN in [1] and CMPNN in [2]. It would be good if the author can do an extra experiment to apply the Hungarian attention module to these two modules. Also, the authors may want to add some discussion about these two papers in the related works section (both papers do have published their codes).

In the current experiment settings, both visual and geometric feature is used. Is it possible for the module to only using geometric features as [1] and [2]?


[1] Wang, Yue, and Justin M. Solomon. "Deep Closest Point: Learning Representations for Point Cloud Registration.", ICCV 2019,
[2] Zhen Zhang, and Wee Sun Lee. "Deep Graphical Feature Learning for the Feature Matching Problem.", ICCV 2019


**Experience Assessment:**

I have published in this field for several years.

**Review Assessment: Checking Correctness Of Derivations And Theory:**

I carefully checked the derivations and theory.

**Review Assessment: Checking Correctness Of Experiments:**

I carefully checked the experiments.

**Review Assessment: Thoroughness In Paper Reading:**

I read the paper thoroughly.

---

> ### Author Response · Authors · 2019-11-13
> **response to review#1**
>
> We thank the reviewer for his constructive suggestion and comments.
>
> As suggested by the reviewer, we have tested Hungarian attention on the model in [2]. To this end, we perform Hungarian algorithm on the output of [2] and establish the attention link during training stage (on synthetic training data). The rest of the settings follow [2]. Results on Pascal-PF dataset can be summarized:
>
> Method                   [2]*             [2]	        [2]+Hungarian attention*
> w/o rotate		88.3		     88.5		89.1
> w     rotate             69.6             69.9            70.2
> where '[2]' refers to the results reported in [2] and '[2]*' corresponds to our implementation using the public code online. We can conclude that the Hungarian Attention can consistently enhance the performance as claimed in our paper. We will add this part to the appendix.
>
> Besides, we are still training our method on geometric features. We will update our paper once ready.
>
> Thanks again to the reviewer for pointing out these two highly relevant papers. We will discuss the methods in [1,2] in the final version.
>
> [1] Wang, Yue, and Justin M. Solomon. "Deep Closest Point: Learning Representations for Point Cloud Registration.", ICCV 2019,
> [2] Zhen Zhang, and Wee Sun Lee. "Deep Graphical Feature Learning for the Feature Matching Problem.", ICCV 2019

---

### Official Review · AnonReviewer4 · 2019-11-02
**Official Blind Review #4**

**Rating:** 6

**Review:**

This work is expressed clearly and well written.

The authors propose a new method to learn graph matching. It contributes in two aspects: 1) a new edge embedding strategy and 2) Hungarian attention incorporating with the loss function. A set of experiments as well as ablation studies have been conducted to show the effectiveness of the method.

However, my concerns are:
1) Is the graph matching algorithms only applied in the field of image matching？How about other fields？
2) And if not, is it more convincing to conduct more experiments across other related fields?
3) It will be better to give algorithm complexity and parameter analysis with the state-of-the-art algorithms since many additional operations are added.

**Experience Assessment:**

I do not know much about this area.

**Review Assessment: Checking Correctness Of Derivations And Theory:**

I did not assess the derivations or theory.

**Review Assessment: Checking Correctness Of Experiments:**

I assessed the sensibility of the experiments.

**Review Assessment: Thoroughness In Paper Reading:**

I read the paper at least twice and used my best judgement in assessing the paper.

---

> ### Author Response · Authors · 2019-11-13
> **response to review#4**
>
> We thank the kind advice and the response is as follows.
>
> Graph matching is a well-established and fundamental problem in computer science and operation research. Besides vision, it also has wide applications in protein alignment, software quality check, graphics, resource allocation, social network analysis etc. with many application papers using off-the-shelf graph matching techniques.
>
> We mainly conduct experiments under the vision setting in line with the majority of literature in this area. This is because there are several public and well-maintained graph matching datasets and experiment protocol in vision community, which makes the evaluation more direct and convenient.
>
> To further show the benefit of our method, we conduct an extra experiment (PASCAL-PF with only geometric edge features) by incorporating the proposed Hungarian Attention mechanism to the method in [1]. We observed performance enhancement consistently. Please refer to 'response to review#1' for more details.
>
> In general, the extra parameters involved in our algorithm compared to PCA are on the calculation of edge embedding. As our method requires the dimensions of node and edge embeddings to be identical, we generally double the amount of parameters in PCA.
>
> [1] Zhen Zhang, and Wee Sun Lee. "Deep Graphical Feature Learning for the Feature Matching Problem.", ICCV 2019

---

### Decision · Program_Chairs · 2019-12-19

**Decision:**

Accept (Poster)

**Comment:**

This paper proposed a new graph matching approach. The main contribution is a Hungarian attention mechanism, which dynamically generates links in computational graph. The resulting matching algorithm is tested on vision tasks.

The main concern of reviews is that the general matching algorithm is only tested on vision tasks. The authors partially addressed this problem by providing new experimental results with only geometric edge features. Other comments of Blind Review #2 are about some minor questions, which have also been answered by the authors.

Overall, this paper proposed a promising graph match approach and I tend to accept it.